# RETRACTED: Protection against Oxidative Stress-Induced Apoptosis by Fermented Sea Tangle (*Laminaria japonica* Aresch) in Osteoblastic MC3T3-E1 Cells through Activation of Nrf2 Signaling Pathway

**DOI:** 10.3390/foods10112807

**Published:** 2021-11-15

**Authors:** So Young Kim, Hee-Jae Cha, Hyun Hwangbo, Cheol Park, Hyesook Lee, Kyoung Seob Song, Jung-Hyun Shim, Jeong Sook Noh, Heui-Soo Kim, Bae-Jin Lee, Suhkmann Kim, Gi-Young Kim, You-Jin Jeon, Yung Hyun Choi

**Affiliations:** 1Anti-Aging Research Center, Dong-eui University, Busan 47340, Republic of Korea; 14731@deu.ac.kr (S.Y.K.); hbhyun2003@naver.com (H.H.); 14769@deu.ac.kr (H.L.); 2Department of Biochemistry, Dong-eui University College of Korean Medicine, Busan 47227, Republic of Korea; 3Department of Parasitology and Genetics, College of Medicine, Kosin University, Busan 49104, Republic of Korea; hcha@kosin.ac.kr; 4Korea Nanobiotechnology Center, Pusan National University, Busan 46241, Republic of Korea; 5Division of Basic Sciences, College of Liberal Studies, Dong-eui University, Busan 47340, Republic of Korea; parkch@deu.ac.kr; 6Department of Medical Life Science, College of Medicine, Kosin University, Busan 49104, Republic of Korea; kssong@kosin.ac.kr; 7Department of Pharmacy, Mokpo National University, Jeonnam 58554, Republic of Korea; s1004jh@gmail.com; 8Department of Food Science & Nutrition, Tongmyong University, Busan 48520, Republic of Korea; jsnoh2013@tu.ac.kr; 9Department of Biological Sciences, College of Natural Sciences, Pusan National University, Busan 46241, Republic of Korea; khs307@pusan.ac.kr; 10Ocean Fisheries & Biology Center, Marine Bioprocess Co., Ltd., Busan 46048, Republic of Korea; hansola82@hanmail.net; 11Center for Proteome Biophysics and Chemistry, Department of Chemistry, College of Natural Sciences, Institute for Functional Materials, Pusan National University, Busan 46241, Republic of Korea; suhkmann@gmail.com; 12Department of Marine Life Science, Jeju National University, Jeju 63243, Republic of Korea; immunkim@jejunu.ac.kr (G.-Y.K.); youjinj@jejunu.ac.kr (Y.-J.J.)

**Keywords:** fermented sea tangle, osteoblast, ROS, apoptosis, Nrf2/HO-1

## Abstract

The purpose of the present study was to explore the efficacy of fermented extract of sea tangle (*Laminaria japonica* Aresch, FST) with *Lactobacillus brevis* on DNA damage and apoptosis in hydrogen peroxide (H_2_O_2_)-stimulated osteoblastic MC3T3-E1 cells and clarify related signaling pathways. Our results showed that exposure to FST significantly improved cell viability, inhibited apoptosis, and suppressed the generation of reactive oxygen species (ROS) in H_2_O_2_-stimulated cells. In addition, H_2_O_2_ triggered DNA damage in MC3T3-E1 cells was markedly attenuated by FST pretreatment. Moreover, H_2_O_2_-induced mitochondrial dysfunctions associated with apoptotic events, including loss of mitochondrial membrane potential (MMP), decreased Bcl-2/Bcl-2 associated x-protein (Bax) ratio, and cytosolic release of cytochrome *c*, were reduced in the presence of FST. FST also diminished H_2_O_2_-induced activation of caspase-3, which was associated with the ability of FST to protect the degradation of poly (ADP-ribose) polymerase. Furthermore, FST notably enhanced nuclear translocation and phosphorylation of nuclear factor erythroid 2-related factor 2 (Nrf2) in the presence of H_2_O_2_ with concomitant upregulation of heme oxygenase-1 (HO-1) expression. However, artificial blockade of this pathway by the HO-1 inhibitor, zinc protoporphyrin IX, greatly abolished the protective effect of FST against H_2_O_2_-induced MC3T3-E1 cell injury. Taken together, these results demonstrate that FST could protect MC3T3-E1 cells from H_2_O_2_-induced damage by maintaining mitochondrial function while eliminating ROS along with activation of the Nrf2/HO-1 antioxidant pathway.

## 1. Introduction

Reactive oxygen species (ROS) is a critical factor in enhancing bone resorption and reducing bone formation [1,2]. Excessive levels of ROS cause oxidative damage to various organs and contribute to the pathogenesis and progression of several oxidative stress-mediated diseases, including osteoporosis [2,3]. Overproduction of ROS also induces oxidative stress, leading to cell death following damage to macromolecules [4,5]. In addition, ROS accumulation can contribute to mitochondrial dysfunction, and cytochrome *c*, which exists between the inner and outer membranes of mitochondria, is released and caspases are activated to induce apoptosis [5,6]. In fact, accumulated results have revealed that excessive production of ROS can lead to bone loss by promoting apoptosis while inhibiting the differentiation of osteoblasts [1,2]. Therefore, research on the discovery and mechanism of antioxidants to protect osteoblasts from the accumulation of ROS is being actively conducted.

Fermentation using microorganisms or microbial enzymes is the process of increasing beneficial bioactive compounds by altering the molecular structure of organic matter [7,8]. Recently, many studies have reported better health benefits compared to unfermented seaweed. Numerous studies have reported that fermented seaweeds have superior health benefits compared to their nonfermented counterparts [9,10]. Sea tangle (*Laminaria japonica* Aresch), a type of edible brown algae, has long been widely consumed as a food supplement and medicinal use in East Asia including Korea. This seaweed is rich in various lipophilic components, hydrophilic fibers, and minerals, and various pharmacological effects were extensively reported in several previous studies [11,12,13,14]. Interestingly, fermented sea tangle extract (FST) with *Lactobacillus brevis* BJ20 content 5.56% of gamma-aminobutyric acid (GABA) that was 49.5% among total free amino acids [15]. It has been reported that FST was enhanced with contents of alanine, valine, glycine, and leucine by the fermentation process [16]. GABA is distributed naturally in various foods and is a non-protein amino acid that is produced by glutamate decarboxylase that catalyzes the irreversible decarboxylation of L-glutamate to GABA [17,18]. In a previous study, GABA-enriched FST had a greater antioxidant effect than unfermented sea tangle in oxidative stress-mediated liver damage [19]. Kang et al. [20] also suggested that administration of FST increased the antioxidant activity in human clinical trials and that FST could be useful as a functional food ingredient. They speculated that the increased antioxidant potential of FST was due to the conversion of glutamate contained in sea tangle to GABA through fermentation by *L. brevis.* [21,22]. Furthermore, Park and Han [23] reported that the butanol fraction of FST protected kidney cells from oxidative damage by inhibiting lipid peroxidation and increasing antioxidant enzyme activity and glutathione concentration. In addition, the application of FST is attracting attention as a potential tool to improve physiological functions such as memory improvement, anti-inflammatory, obesity reduction, and stress management [21,22,24,25,26]. Recently, we reported that FST inhibited osteoclastogenesis, which was due to the blockade of oxidative stress involved in the activation of the nuclear factor erythroid 2-related factor 2 (Nrf2) pathway [27]. However, the effects and mechanisms of FST in oxidative stress-induced osteoblast dysfunction remain unclear.

Therefore, we investigated the protective effect of FST on oxidative stress-mediated cellular dysfunction with hydrogen peroxide (H_2_O_2_) in MC3T3-E1 osteoblasts and explore the related signaling pathways [28,29].

## 2. Materials and Methods

### 2.1. MC3T3-E1 Cell Culture 

MC3T3-E1 cells were purchased from the American Type Culture Collection (Manassas, VA, USA) and maintained in minimum essential medium (α-MEM) supplemented with 1% antibiotics and 10% fetal bovine serum (all from WelGENE Inc., Gyungsan, Korea) at 37 °C in an atmosphere containing 5% CO_2_. FST, a fermented sea tangle extract using *L. brevis* [22], was provided by Marine Bioprocess Co., Ltd. (Busan, Korea). FST and H_2_O_2_ (Sigma-Aldrich Chemical Co., St. Louis, MO, USA) were diluted with α-MEM and distilled water.

### 2.2. Cell Viability

Cell viability was measured using a 3-(4,5-dimethylthiazol-2-yl)-2,5-diphenyltetrazolium bromide (MTT; Sigma-Aldrich Chemical Co.) assay [30]. The absorbance at 540 nmzwas detected using a microplate reader (Molecular Devices, Sunnyvale, CA, USA).

### 2.3. Analysis of ROS Content

Intracellular ROS levels were investigated using 2′7′-dichlorodihydrofluorescein diacetate (DCF-DA; Sigma-Aldrich Chemical Co.) dye as previously described [31]. The fluorescence intensity was observed using a flow cytometer (Becton Dickinson, San Jose, CA, USA) and a fluorescence microscope (Carl Zeiss, Oberkochens, Germany) 

### 2.4. Western Blot Analysis

For expression analysis of target proteins by immunoblotting, whole proteins were isolated according to published method [30]. Cytoplasmic and mitochondrial proteins were extracted using a mitochondrial fractionation kit (Active Motif, Inc., Carlsbad, CA, USA). The same amount of protein was separated by SDS-PAGE, electrophoresis, and transferred. The membranes were incubated with 5% skim milk then incubated with primary antibodies and correlated secondary antibodies (Appendix A). The membranes were exposed enhanced chemiluminescence (Amersham Biosciences, Westborough, MA, USA) and visualized under a Fusion FX Image system (Vilber Lourmat, Torcy, France). Densitometric analysis of the bands was performed using the ImageJ^®^ software (v1.48, NIH, Bethesda, MD, USA).

### 2.5. Comet Assay

To evaluate DNA damage, we used the Comet Assay^®^ kit (Trevigen Inc., Gaithersburg, MD, USA), following the manufacturer’s instructions [32].

### 2.6. Mitochondrial Membrane Potential (MMP, Δψm) Assay

5,5′6,6′-tetrachloro-1,1′,3,3′-tetraethyl-imidacarbocyanine iodide (JC-1; Sigma-Aldrich Chemical Co.) staining was performed to assess MMP, as according to the manufacturer’s procedure [33].

### 2.7. Apoptosis Analysis

To investigate cell death mode, Annexin V-fluorescein isothiocyanate (FITC) and propidium iodide (PI) double staining (Becton Dickinson, San Jose, CA, USA) was performed following published procedures [34]. Quantity of annexin V-positive cells, apoptotic cells, was determined by flow cytometry.

### 2.8. Immunofluorescence Staining for p-Nrf2

To observe intracellular expression of p-Nrf2, cells cultured on glass coverslips were pretreated with or without FST for 1 h and additionally exposed to H_2_O_2_ for 24 h. After fixation and permeabilization, the cells were incubated anti-rabbit p-Nrf2 antibody (Abcam, Inc., Cambridge, UK), and then proved with Alexa Fluor 647-conjugated secondary antibody (Abcam, Inc.). Additionally, nuclei were counterstained with 4′,6′-diamidino-2-phenylindole (DAPI; Sigma-Aldrich Chemical Co.). The stained cells were observed under a fluorescence microscope as previously described [35].

### 2.9. Antioxidant Capacity and Total Phenolic Content

The 2,2-diphenyl-1-picrylhydrazyl free radical (DPPH^●^) scavenging capacity of FST was measured as previously described [36]. The 2,20-azino-bis (3-ethyl benzothiazoline-6-sulfonic acid (ABTS^●^^+^) assay was based on the method of Du et al. [37]. In DPPH and ABTS assay, trolox (0.1 mg/mL) was used as positive control. Total phenolic content (TPC) was determined using the Folin–Ciocalteu reagent method [38]. The results were calculated as milligrams of gallic acid equivalent (GAE) per 100 g.

### 2.10. Statistical Analysis

All experiments were repeated three times. The data are expressed as mean and standard deviation (SD). One-way ANOVA was performed to determine significant differences (*p* < 0.05) using SPSS 25.0 (SPSS Inc., Chicago, IL, USA).

## 3. Results

### 3.1. FST Prevents the Reduction of Viability of MC3T3-E1 Cells Caused by H_2_O_2_ Treatment

To investigate the effect of FST on cytotoxicity in H_2_O_2_-stimulated MC3T3 cells, cell viability after treatment with FST and H_2_O_2_ alone was first measured using the MTT assay. Figure 1A showed that FST has no cytotoxicity up to 800 μg/mL in MC3T3 cells, but a slightly cytotoxic effect at 1000 μg/mL. On the other hand, cell viability was gradually decreased with the increasing concentration of H_2_O_2_ used for treatment. Cells treated with 200 μM H_2_O_2_ showed the viability of about 60% (Figure 1B). Therefore, the treatment concentration of H_2_O_2_ to suppress cell viability was selected to be 200 μM and the maximum concentration of FST to investigate its protective effect was set to be 800 μg/mL. Figure 1C showed that FST markedly attenuated the decreasing viability by H_2_O_2_ in a concentration-dependent manner (Figure 1C).

### 3.2. FST Inhibits ROS Generation Induced by H_2_O_2_ in MC3T3-E1 Cells

To investigate whether H_2_O_2_-induced cytotoxicity was related to oxidative stress and whether FST could block it, the degree of ROS generation was determined. Flow cytometry results showed that ROS accumulation was greatly increased in H_2_O_2_-treated MC3T3-E1 cells (Figure 2A,B). Consistent with this result, the images of a fluorescence microscope were markedly increased by H_2_O_2_ (Figure 2C). However, the accumulation of ROS increased by H_2_O_2_ was remarkably decreased by FST. The result suggested that H_2_O_2_-induced cytotoxicity in MC3T3-E1 cells was involved in ROS accumulation and that the protective effect of FST against H_2_O_2_ was related to its antioxidant activity.

### 3.3. FST Diminishes DNA Damage Induced by H_2_O_2_ in MC3T3-E1 Cells

The effect of FST against H_2_O_2_-induced DNA damage was assessed. Figure 2D,E showed the expression level of phosphorylated nuclear histone H2A.X protein (γH2A.X) was strongly up-regulated in H_2_O_2_-treated cells. However, its expression was not induced by FST. Additionally, migration of damaged DNA fragments by electrophoresis was distinctly observed following by H_2_O_2_. However, these DNA tails were not generated in cells pretreated with FST (Figure 2F). The result suggested that the protective effect of FST against DNA damage caused by H_2_O_2_ might be contributed to the inhibition of ROS production.

### 3.4. FST Suppresses Mitochondrial Dysfunction in H_2_O_2_-Stimulated MC3T3-E1 Cells

We next investigated changes in MMP using JC-1 staining to evaluate whether FST-mediated ROS suppression and DNA repair were correlated with blockade of mitochondrial dysfunction. As shown in Figure 3A,B, the loss of MMP was significantly enhanced in H_2_O_2_-stimulated cells. Moreover, the expression of cytochrome *c* in the cytoplasm was increased following H_2_O_2_, while its expression in the mitochondria was decreased (Figure 3C,D). However, these changes caused by H_2_O_2_ were completely blocked by FST, suggesting that FST effectively inhibited H_2_O_2_-mediated loss of MMP to prevent oxidative stress-induced mitochondrial damage.

### 3.5. FST Attenuates H_2_O_2_-Induced Apoptosis in mc3t3-e1 Cells

We further assessed whether FST could prevent H_2_O_2_-induced apoptosis. Figure 4A,B showed that the population of annexin V-positive cells, which means apoptosis-inducing cells, was greatly increased by H_2_O_2_ treatment, but was significantly decreased in the presence of FST, indicating that FST attenuated H_2_O_2_-induced apoptosis in MC3T3-E1 cells. To identify the mechanism involved in the protective effect of FST on H_2_O_2_-induced apoptosis, the effects of FST on the expression of the Bcl-2 family and caspase-3 were investigated. Figure 4C,D showed that the expression of Bcl-2 protein was decreased by H_2_O_2_, whereas the expression of Bcl-2 associated x-protein (Bax) was unchanged by H_2_O_2_. In addition, the expression of inactive pro-forms of caspase-3 was down-regulated and poly (ADP-ribose) polymerase (PARP) was degraded by H_2_O_2_ treatment. However, all these alterations were entirely blocked by FST pretreatment.

### 3.6. FST Stimulates the Nrf2/HO-1 Signaling Pathway in H_2_O_2_-Stimulated MC3T3-E1 Cells

Next, we evaluated whether the Nrf2 signaling pathway was associated with the antioxidant activity of FST. As indicated in Figure 5A,B, Nrf2 expression was partially increased in FST alone or H_2_O_2_ alone-treated cells. However, its expression was markedly increased when FST and H_2_O_2_ were treated together. The phosphorylation of Nrf2 was slightly up-regulated by H_2_O_2_ treatment, while the expression was markedly enhanced by FST and H_2_O_2._ In addition, the expression of heme oxygenase-1 (HO-1), a representative downstream regulator of Nrf2, was improved in the co-treatment cells. Furthermore, through immunofluorescence experiments, it was confirmed that p-Nrf2 was uniquely expressed in the nucleus rather than in the cytoplasm of cells co-treated with FST and H_2_O_2_ (Figure 5C). These results indicate that Nrf2 could be transferred to the nucleus after phosphorylation to act as a transcription factor to induce the transcriptional activity of HO-1 in the co-treatment cells with FST and H_2_O_2_.

### 3.7. Protection against H_2_O_2_-Induced Cytotoxicity by FST Is Involve in Stimulation of the Nrf2/HO-1 Signaling Pathway in MC3T3-E1 Cells

In order to elucidate the contribution of the Nrf2/HO-1 signaling pathway in the benefit of FST against the H_2_O_2_-induced cytotoxic effect, MC3T3-E1 cells were pretreated with the HO-1 antagonist, ZnPP, and then incubated with FST and H_2_O_2_. As shown in Figure 6A,B, the inhibitory effect of FST on H_2_O_2_-induced ROS generation was significantly reduced in the presence of ZnPP. At the same time, in the presence of ZnPP, the inhibitory efficacy of FST on the suppression of expression of Bcl-2 and pro-caspase-3, cleavage of PARP and cytochrome *c* leakage from mitochondria to the cytoplasm were also abrogated (Figure 6C–F). The suppressive efficacy of FST on H_2_O_2_-mediated apoptosis was also abolished by ZnPP (Figure 7). These results suggest the notion that FST protects against oxidative stress-induced cellular damage by stimulation of the Nrf2/HO-1 pathway in MC3T3-E1 cells.

### 3.8. Antioxidant Capacity and Total Phenolic Contents of FST

To evaluate whether FST directly eliminate the active oxygen, we performed DPPH and ABTS assay. Figure 8A showed that the DPPH radical scavenging capacity of FST showed high activity to approximately 50% from 0.015 mg/mL concentration, and saturated over 1 mg/mL. Furthermore, another stable free radical cation, ABTS, was also markedly eliminated by FST, and the half-maximal inhibitory concentration (IC_50_) value was 0.307 mg/mL (Figure 8B). Meanwhile, 0.1 mg/mL of Trolox, a positive control, perfectly eliminated DPPH and ABTS radicals. In addition, we further measured the TPC of FST, and the result showed that the TPC of FST was 63.32 mg GAE/g (Table 1).

## 4. Discussion

Herein, we evaluated the efficacy of FST on oxidative stress-enhanced cellular damage and apoptotic cell death using H_2_O_2_ to mimic oxidative stress in MC3T3-E1 osteoblastic cells. The present findings showed that FST markedly blocked H_2_O_2_-induced cytotoxicity, which was due to its ROS scavenging activity. We also demonstrated that the antioxidant activity of FST was accompanied by activation of the Nrf2/HO-1 pathway. 

Because oxidative stress causes oxidative injury to various intracellular macromolecules, including nucleic acids, failure to restore sustained oxidative stress can eventually induce apoptosis, resulting in tissue and organ damage [6,39]. In particular, in osteoblasts sensitive to oxidative stress, mitochondrial damage is induced by excessive ROS generation, which negatively affects osteogenic function [1,2]. In addition, mitochondrial damage serves as an initiation signal for activation of the intrinsic apoptosis pathway, eventually leading to osteoblast damage through induction of apoptosis [2,40]. Oxidatively damaged DNA also induces genetic mutations and disrupts the homeostasis of osteoblast, leading to osteogenic damage as well as failure to induce osteoblast differentiation [41,42]. In this study, we found that FST had strong antioxidant activity by completely blocking the generation of ROS by H_2_O_2_. Therefore, we investigated the efficacy of FST on H_2_O_2_-induced DNA injury by performing comet assay, a method to assess DNA damage in individual cells induced by oxidative stress [43], and by analyzing the γH2A.X, an indicator of double-strand break [44]. As a result, we found that FST effectively suppressed comet tail length and the over-expression of γH2A.X by H_2_O_2_, suggesting that FST significantly blocked DNA damage to oxidative stress.

Mitochondria are highly susceptible to oxidative stress [45,46,47]. Mitochondrial membrane depolarization is a classic feature when mitochondria are damaged. Therefore, depolarization of MMP is a hallmark of disruption of mitochondrial membrane integrity [48,49]. According to previous studies, the induction of apoptosis in osteoblasts by H_2_O_2_ was due to the release of cytochrome *c* into the cytoplasm caused by disrupted mitochondrial membrane stability [29,50,51]. Cytochrome *c* promotes the activity of effector caspases in the cytoplasm to complete apoptosis [52,53]. In this study, the level of MMP reduced by H_2_O_2_ treatment was maintained at control levels in FST-treated cells. Subsequently, cytochrome *c* expression was predominantly expressed in the cytoplasm in H_2_O_2_-treated cells. However, its expression was counteracted by pretreatment with FST, suggesting that mitochondrial integrity was maintained by pretreatment with FST. Moreover, the reduced expression of the Bcl-2 protein by H_2_O_2_ was reversed by FST pretreatment. Additionally, the inactive form of caspase-3 was decreased by H_2_O_2_, indicating that caspase-3 was activated by H_2_O_2_, which was canceled in the presence of FST. This was linked to inhibited cleavage of PARP by blocking the activity of caspase-3 by FST. Because Bcl-2 family proteins control the permeability of the mitochondrial outer membrane, thereby regulating the release of cytochrome *c* from the mitochondria into the cytoplasm [52,53], it is presumed that the increment of Bcl-2/Bax ratio by FST plays a key role in suppressing the H_2_O_2_-mediated apoptosis. Therefore, FST could be a potential antioxidant to suppress DNA damage and apoptosis by inhibiting ROS accumulation, which results in mitochondria-mediated apoptosis pathways in MC3T3-E1 cells.

A number of intracellular signaling molecules and enzymes are required for the maintenance of redox homeostasis for defense against oxidative stress. Among them, Nrf2 controls the transcriptional activity of phase II antioxidant enzymes by binding to antioxidant response elements (AREs) in response to oxidative stress [54,55]. Nrf2 is tightly regulated by its negative regulator, E3 ligase adapter Kelch-like ECH-associated protein 1 (Keap1) under physiological conditions. However, under pathological conditions, such as overproduction of ROS, Nrf2 must be released from Keap1 in order to translocate to the nucleus [55,56]. However, for Nrf2 to be transferred to the nucleus and function as a transcription factor, its phosphorylation must precede [54,55]. Thus, the nuclear translocation after phosphorylation of Nrf2 implies that Nrf2 is activated for transcriptional activation of downstream enzymes, including HO-1, by binding to the AREs. Recently, we demonstrated that the activation of Nrf2 by FST contributed to the inhibition of osteoclastogenesis through blockade of oxidative stress [27]. Supporting this result, our data showed that FST could further activate Nrf2 in the presence of H_2_O_2_, as evidence for the predominant nuclear expression of p-Nrf2 in H_2_O_2_ and FST-cotreated cells. Therefore, the up-regulation of HO-1 is probably inferred as a result of the activation of Nrf2. HO-1, a stress-inducible and redox-sensitive enzyme, contributes to the inhibition of oxidative stress by breaking down heme into its metabolites biliverdin, carbon monoxide, and iron [55,57]. In this regard, we used ZnPP, to determine whether the efficacy of FST against oxidative stress was involved in the HO-1 signaling pathway. Intriguingly, ZnPP significantly reversed H_2_O_2_-induced apoptosis by reducing ROS generation, upregulating the anti-apoptotic protein Bcl-2, decreasing cytochrome *c* release into the cytoplasm, and attenuating apoptosis cascade activation (caspase-3 and PARP). Additionally, ZnPP also abrogated the protective effect of FST against H_2_O_2_-mediated cytotoxicity. Overall, these results revealed that FST protected MC3T3-E1 cells from H_2_O_2_-induced apoptosis through activating the Nrf2/HO-1 signaling pathway (Figure 9). To verify whether the FST-induced Nrf2/HO-1 pathway was involved directly in the elimination of active oxygen, we evaluated the antioxidant capacity by DPPH assay, ABTS assay and Folin–Ciocalteu tests. The ABTS radical removes a hydrogen atom transfer by an antioxidant, while the DPPH radical removes a single electron transfer by an antioxidant [58]. Folin–Ciocalteu tests are also based on the single electron transfer mechanism [58]. In the present study, our finding showed that FST has a strongly scavenging activity of free radicals, including DPPH and ABST radicals, as well as a high content of total phenol. Based on this result, we suggested that the FST-induced Nrf2/HO-1 pathway directly involves the elimination of active oxygen, resulting in antioxidant activity.

## 5. Conclusions

In summary, our finding showed that FST significantly suppressed DNA damage and apoptosis by reducing ROS generation in MC3T3-E1 osteoblasts exposed to H_2_O_2_. In addition, we found that the apoptosis-blocking effect of FST was associated with improved mitochondrial function by regulating Bcl-2 family proteins and inhibiting the activation of caspase cascade through the suppression of the cytosolic release of cytochrome *c*. Furthermore, our results indicated that the Nrf2/HO-1 pathway contributed at least to the antioxidant activity of FST. Therefore, FST has excellent applicability as a natural substance with the potential to inhibit oxidative stress-mediated osteoblast dysfunction. However, further studies on the role of FST in the activation of Nrf2-dependent regulators other than HO-1 and upper signaling pathways involved in Nrf2 phosphorylation are needed. In addition, verification of their effectiveness, and follow-up studies through an Nrf2 knock-out mice experiment should be conducted.

## Figures and Tables

**Figure 1 foods-10-02807-f001:** Effect of fermented sea tangle extract (FST) on hydrogen peroxide (H_2_O_2_)-induced cytotoxicity of MC3T3-E1 cells. (**A**,**B**) Cells were treated with FST or H_2_O_2_ for 24 h (**A**,**B**). (**C**) Cells were treated with or without FST for 1 h prior to exposure to (200 μM H_2_O_2_ for 24 h. Cell viability was presented as the survival rate of cells relative to that of the untreated control cells. * *p* < 0.05, ** *p* < 0.01 *** *p* < 0.001 vs. control cells. ^#^ *p* < 0.05, ^##^ *p* < 0.01 vs. H_2_O_2_-treated cells.

**Figure 2 foods-10-02807-f002:** The protective effect of FST on reactive oxygen species (ROS) generation and DNA damage in H_2_O_2_-stimulated MC3T3-E1 cells. Cells were pre-treated with or without FST for 1 h and exposed to H_2_O_2_ for 1 h (**A**–**C**) or 24 h (**D**,**E**). (**A**–**C**) Fluorescence intensities of 2′7′-dichlorodihydrofluorescein diacetate (DCF-DA) were determined by flow cytometry (**A**,**B**) or fluorescence microscopy (**C**). (**B**) Ratios of DCF-positive cells were statistically quantified. ** *p* < 0.01 and *** *p* < 0.001 vs. control cells; ^###^ *p* < 0.001 vs. H_2_O_2_-treated cells. (**D**) The expression of rH2AX. Actin was loading control. (**E**) Relative band density of rH2AX. *** *p* < 0.001 vs. control cells. ^###^ *p* < 0.001 vs. H_2_O_2_-treated cells. (**F**) DNA damage was also detected by a comet assay. Representative images are shown.

**Figure 3 foods-10-02807-f003:** Effect of FST on mitochondrial damages in H_2_O_2_-stimulated MC3T3-E1 cells. Cells were pre-treated with or without FST (800 μg/mL) and exposed to H_2_O_2_ (200 μM) for 24 h. (**A**,**B**) Mitochondrial membrane potential (MMP) was measured by flow cytometry after 5,5′6,6′-tetrachloro-1,1′,3,3′-tetraethyl-imidacarbocyanine iodide (JC-1) staining. (**A**) Representative profiles are shown. (**B**) Ratios of JC-1 aggregates to monomers are expressed. ** *p* < 0.01 and *** *p* < 0.001 vs. control cells. ^###^ *p* < 0.001 vs. H_2_O_2_-treated cells. (**C**) The expression of cytochrome c in mitochondrial (M.F.) and cytoplasmic fractions (C.F.). Cytochrome *c* oxidase subunit IV (COX IV) and actin were analyzed as internal controls for mitochondrial and cytosolic fractions, respectively. (**D**) Relative band density. *** *p* < 0.001 vs. control cells. ^###^ *p* < 0.001 vs. H_2_O_2_-treated cells.

**Figure 4 foods-10-02807-f004:** Protective effect of FST on apoptosis in H_2_O_2_-stimulated MC3T3-E1 cells. Cells were pre-treated with or without FST (800 μg/mL) and exposure to H_2_O_2_ (200 μM) for 24 h. (**A**,**B**) Cells were stained with annexin V and propidium iodide (PI) and then analyzed by flow cytometry. (**A**) Percentages of apoptotic cells are shown as annexin V-positive cells. (**B**) Quantitative analysis of apoptotic cells in percentage. *** *p* < 0.001 vs. control cells. ^##^ *p* < 0.01 and ^###^ *p* < 0.001 vs. H_2_O_2_-treated cells. (**C**) The expression of apoptosis regulators. Actin was used for loading control. (**D**) Relative band density of apoptosis regulators. ** *p* < 0.01 and *** *p* < 0.001 vs. control cells. ^###^ *p* < 0.001 vs. H_2_O_2_-treated cells.

**Figure 5 foods-10-02807-f005:** Activation of nuclear factor erythroid 2-related factor 2/ heme oxygenase-1 (Nrf2/HO-1) signaling pathway by FST in H_2_O_2_-treated MC3T3-E1 cells. Cells were pre-treated with or without FST (800 μg/mL) and exposed to H_2_O_2_ (200 μM) for 24 h. (**A**) The expression of Nrf2, phosphorylated (p)-Nrf2, and HO-1. Actin was used to loading control. (**B**) Relative band density expressed as fold of control cells. * *p* < 0.05, ** *p* < 0.01 and *** *p* < 0.001 vs. control cells. ^#^ *p* < 0.05 and ^###^ *p* < 0.001 vs. H_2_O_2_-treated cells. (**C**) Immunofluorescence staining ofp-Nrf2 (green), 4′,6′-diamidino-2-phenylindole (DAPI, blue) staining for nuclear staining. Representative micrographs are shown. Scale bar; 100 μm.

**Figure 6 foods-10-02807-f006:** Effect of zinc protoporphyrin IX (ZnPP) on FST-mediated attenuation of MC3T3-E1 cells against H_2_O_2_-induced ROS production and changes in expression of apoptosis regulators. Cells were pretreated with the HO-1 antagonist ZnPP (5 μM) with or without FST (800 μg/mL) for 1 h, and then subjected to treatment with H_2_O_2_ (200 μM) for 1 h (**A**,**B**) or 24 h (**C**,**D**). (**A**) Intracellular ROS levels were determined using DCF-DA dye. (**B**) Ratios of DCF-positive cells were statistically quantified (**C**) The expression of apoptosis regulators. Actin was used to loading control. (**E**) Expression of cytochrome *c* in mitochondrial and cytoplasmic fractions. COX IV and actin were analyzed as internal controls for mitochondrial and cytosolic fractions, respectively. (**D**,**F**) Relative band density. (**B**,**D**,**F**) * *p* < 0.05, ** *p* < 0.01 and *** *p* < 0.001, and *** *p* < 0.001 vs. control cells. ^#^ *p* < 0.05, ^##^ *p* < 0.01 and ^###^ *p* < 0.001 vs. H_2_O_2_-treated cells. ^$^ *p* < 0.05, ^$$^
*p* < 0.01 and ^$$$^ *p* < 0.001 vs. H_2_O_2_ and FST-treated cells.

**Figure 7 foods-10-02807-f007:** Effect of ZnPP on FST-induced protection of apoptosis and cytotoxicity by H_2_O_2_ in MC3T3-E1 cells. Cells were pretreated with the ZnPP (5 μM) with or without FST (800 μg/mL) for 1 h and then subjected to treatment with H_2_O_2_ (200 μM) for 24 h. (**A**,**B**) Cells were double-stained with annexin V /PI and then analyzed by flow cytometry. (A) Percentages of apoptotic cells are shown as annexin V-positive cells. (**B**) Quantitative analysis of apoptotic cells. (**C**). Cell viability. (**B**,**C**) *** *p* < 0.001 vs. control cells. ^###^ *p* < 0.001 vs. H_2_O_2_-treated cells. ^$$^ *p* < 0.01 and ^$$$^ *p* < 0.001 vs. H_2_O_2_ and FST-treated cells.

**Figure 8 foods-10-02807-f008:** Antioxidant capacity of FST. (**A**) DPPH radical scavenging activity of FST. (**B**) ABTS radical scavenging activity of FST (**B**). Trolox (0.1 mg/mL) was used as the positive controls. Data are represented the mean ± standard deviation of two independent experiments.

**Figure 9 foods-10-02807-f009:** Scheme summarizing the protective effect of FST against oxidative stress-induced MC3T3-E1 osteoblast apoptosis by inhibiting ROS-dependent mitochondrial dysfunction and activating Nrf2/HO-1 antioxidant response pathway.

**Table 1 foods-10-02807-t001:** Total phenolic content (TPC) of FST.

Sample	TPC (mg/GAE/g)
FST	60.32 ± 0.22

Data are represented as mg gallic acid equivalent (GAE) per mg.

## Data Availability

The original contributions presented in the study are included in the article/Appendix A. Further inquiries can be directed to the corresponding author/s.

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
