# Peer review of "RETRACTED: Protection against Oxidative Stress-Induced Apoptosis by Fermented Sea Tangle (*Laminaria japonica* Aresch) in Osteoblastic MC3T3-E1 Cells through Activation of Nrf2 Signaling Pathway"

_foods, 2021, doi:10.3390/foods10112807_

Round 1
Reviewer 1 Report
The work by Kim et al. aims to demonstrate cytoprotective properties of a fermented sea tangle extract in the osteoblastic MC3T3-E1 cell line after exposure to hydrogen peroxide. The authors link their findings to an activation of the Nrf2/HO-1 signaling pathway.
There are some concerns about the manuscript:
- It does not become really clear what the sea tangle extract is composed of and what would be the pharmacologically active compounds leading to an activation of Nrf2 and downstream effectors. Please include a concise description of how the extract was manufactured and, if possible, what it contains.
- In the methods section, the list of reagents is quite confusing. The authors should clearly indicate relevant reagents, antibodies etc. with the description of the respective methods. Merely stating that antibodies were obtained from Santa Cruz is rather not helpful…
- All western blots are missing molecular weight standards. These must be included. Furthermore, quantifications of western blots should be included.
- Detection of Nrf2 is a tricky business. As no molecular weight standards were included it is difficult to judge whether the correct lanes were assessed.
- In fig. 5 b, scale bar should be included.
Author Response
Dear reviewer,
Thank you for the opportunity to revise our manuscript “Protection against Oxidative Stress-induced Apoptosis by Fermented Sea Tangle (Laminaria japonica Aresch) in Osteoblastic MC3T3-E1 Cells through Activation of Nrf2 Signaling Pathway”. We appreciate the careful review and constructive suggestions. It is our belief that the manuscript is substantially improved after making the suggested edits. Following this letter are the reviewers’ comments with our responses in characters with blue color, including how and where the text was modified. To make the changes easier to identify where necessary in the text, we have marked them using red pen.
The revision has been developed in consultation with all coauthors, and each author has given approval to the final form of this revision. Thank you again for giving us the opportunity to revise and resubmit this manuscript.
Sincerely,
Yung Hyun Choi, Ph.D.

Reviewer 2 Report
FST can suppress DNA damage and apoptosis by reducing ROS generation in MC3T3-E1 osteoblasts exposed to H2O2. These results are seen in MC3T3-E1 osteoblast, is these findings can be seen in other cell or tissues. This should be mentioned in study limitation.
Alternative pathway for apoptosis may need to discussed if further experiments conducted.
Author Response

(The authors gave the same response as above.)

Reviewer 3 Report
The authors described the protective effects of fermented sea tangle (FST) against hydrogen peroxide induce damage in osteoblastic cells. Fermented foods usually have many healthy functions and it is interesting that FST act as a trigger of Nrf2 signaling. The authors used a conventional experimental system, namely H2O2 addition on osteoblast, and obtained many data from stress induced Nrf2 activation to mitochondrial damages.
However, several experiments are necessary for understanding of the function of FST.
At first, it is necessary to check the active oxygen elimination ability of FST. It may contain some elimination compounds.
Authors described FST activate Nrf2 pathway. However, FST alone do not phosphorylate Nrf2 in Figure 5. It seemed that some oxidated compounds of FST by H2O2 activated Nrf2.
FST might have some anti-oxidative activities, but it must to be carefully construct the experiments.
Author Response

(The authors gave the same response as above.)

Round 2
Reviewer 2 Report
The manuscript is well revised. Further experiment may be conducted in future to support their theory for real-world application about FST protection on oxidative injury, that is "FST could protect MC3T3-E1 cells from H2O2-induced dam-44 age by maintaining mitochondrial function while eliminating ROS along with activation of the 45 Nrf2/HO-1 antioxidant pathway"
Reviewer 3 Report
The revised manuscript is sufficiently interesting to readers. The results have been improved. The conclusion was changed to a reasonable explanation of the effects of FST.